# What Predictability for Animal Models of Peripheral Vestibular Disorders?

**DOI:** 10.3390/biomedicines10123097

**Published:** 2022-12-01

**Authors:** Brahim Tighilet, Jessica Trico, Frédéric Xavier, Christian Chabbert

**Affiliations:** 1Aix Marseille Université-CNRS, Laboratoire de Neurosciences Cognitives, LNC UMR 7291, 13331 Marseille, France; 2GDR Vertige CNRS Unité GDR2074, France

**Keywords:** vestibular syndrome, animal models, predictability

## Abstract

The different clinical entities grouped under the term peripheral vestibulopathies (PVs) or peripheral vestibular disorders (PVDs) are distinguished mainly based on their symptoms/clinical expression. Today, there are very few commonly accepted functional and biological biomarkers that can confirm or refute whether a vestibular disorder belongs to a precise classification. Consequently, there is currently a severe lack of reliable and commonly accepted clinical endpoints, either to precisely follow the course of the vertigo syndrome of vestibular origin or to assess the benefits of therapeutic approaches, whether they are pharmacological or re-educational. Animal models of PV are a good means to identify biomarkers that could subsequently be exploited in human clinical practice. The question of their predictability is therefore crucial. Ten years ago, we had already raised this question. We revisit this concept today in order to take into account the animal models of peripheral vestibular pathology that have emerged over the last decade, and the new technological approaches available for the behavioral assessment of vestibular syndrome in animals and its progression over time. The questions we address in this review are the following: are animal models of PV predictive of the different types and stages of vestibular pathologies, and if so, to what extent? Are the benefits of the pharmacological or reeducational therapeutic approaches achieved on these different models of PV (in particular the effects of attenuation of the acute vertigo, or acceleration of central compensation) predictive of those expected in the vertiginous patient, and if so, to what extent?

## 1. Notion of “Predictability” Applied to Peripheral Vestibulopathies

The notion of predictability of a study model in the field of experimental biology refers to its ability to predict and/or reproduce as closely as possible in vivo, in vitro or in silico biological events and functional correlates which will be found in humans. In the case of study models of vestibular disorders, it will thus be a question of predicting, on the basis of experimental observations, the biological and functional consequences of the administration of a pharmacological compound of interest or of a rehabilitation approach [1].

This notion is of great importance in the case of peripheral vestibulopathies referenced in the International Classification of Diseases under the codes (H81.0-MD-, H81.1-BPPV-, H81.2-VN-, H81.3 other peripheral vertigos, H81.8, and H81.9 other PVD; [2]). These pathologies are characterized by sudden episodes of functional alterations, composed of static and dynamic disorders, which may include loss of postural balance, locomotion disorders, spatial disorientations, vestibulo-ocular reflex deficits causing the expression of nystagmus, and alterations in vegetative and cognitive functions [3]. The symptoms gradually subside, each with its own kinetics, generally leading to an almost total disappearance of the syndrome. This phenomenon of behavioral recovery is referred as “vestibular compensation” [4].

One of the common characteristics of peripheral vestibulopathies is that in the majority of cases, the causes of the pathology are not clearly identified. This results largely from the fact that the sensory organs of the inner ear, because of their confinement within the temporal bone, are not accessible in the patient, either to dosages or to direct electrophysiological recordings, and even very partially to high resolution medical imaging. Thus, apart from certain situations (drug ototoxicity, temporal bone trauma, fistula or vestibular atelectasis), the main listed peripheral vestibulopathies have only supposed etiologies.

Because of this lack of information on their respective etiologies, the different clinical entities grouped under the term peripheral vestibulopathy (PV) or peripheral vestibular disorders (PVDs) are distinguished mainly on the basis of their functional correlates. These symptoms are also very heterogeneous for the same clinical entity, each parameter varying with its own kinetics. In addition, there are currently very few biomarkers measurable by clinical imaging, assay, or electrophysiological recordings, which might make it possible to confirm or refute whether a vestibular disorder corresponds to a particular classification. As a result, there is currently a severe lack of reliable and commonly accepted clinical endpoints for, on the one hand, precisely monitoring the course of the vertigo syndrome, and on the other hand, assessing the benefits of therapeutic approaches, whether pharmacological or re-educational. The recent failures of many clinical trials of drug candidates with antivertigo properties are an illustration of this.

The modern era of the development of animal models of vestibular damage really began with the work of the French physician and biologist PMJ Flourens (1794–1867). He was the first, thanks to his work on pigeons, to establish a direct link between damage to the inner ear vestibular sensors and the characteristic posturo-locomotor deficits encountered in vertigo patients. Researchers therefore set out, on the basis of an epidemiological rationale, to reproduce in animals the potential conditions of aggression of the peripheral vestibular system and to select those inducing the closest symptomatology to that encountered in patients. From the 1960s, the first electrophysiological recordings made at the level of brainstem vestibular nuclei of animals subjected to a unilateral vestibular lesion made it possible to demonstrate that the symptoms constituting the vestibular syndrome result from an electrophysiological imbalance between the intact and deafferented vestibular nuclei, with weak spontaneous electrical activity on the deafferented side, and intense activity on the intact side (Figure 1) [5,6]. On the basis of this observation, several animal models mimicking the neurophysiological situations underlying the vestibular syndrome have since been developed (see below).

## 2. Animal Models of Peripheral Vestibulopathies

Animal models of PV can be classified into different categories depending on whether they mimic putative pathogenic conditions or reproduce neurophysiological situations underlying the vertiginous syndrome. A classification on this basis is proposed below.

### 2.1. Animal Models Mimicking Putative Pathogenic Conditions

#### 2.1.1. Models of Viral or Bacterial Infections

Hearing loss and vestibular damage are well-established comorbidities of viral and bacterial infections [7,8,9]. Several hypotheses have been put forward to explain vestibular pathologies such as the acute unilateral peripheral vestibulopathy (AUPV). The most commonly accepted remains a viral origin. Post-mortem studies revealed the presence of herpes simplex virus type 1 (HSV-1) on the vestibular ganglia [10,11]. However, individually, it is not possible to confirm by serology a concomitant viral infection. The mechanisms by which inner ear sensor damage occurs are not yet clearly established. Different models of viral or bacterial infections of the inner ear have been developed [12,13]. These models, due to the fact that they do not necessarily generate vertigo syndrome, and due to their lack of reproducibility, are not currently part of the arsenal of animal models used in preclinical studies for efficacy tests of drug candidates.

#### 2.1.2. Models of Ototoxic Damage

The term ototoxicity refers to the property of certain compounds of being toxic, in particular with respect to the ear (oto in Latin), which includes both the cochlea and the vestibule. When specifically affecting the vestibule, ototoxicity can cause dizziness and unsteadiness, potentially leading to serious disabilities (for review: [14,15]). Depending on the dose and duration of administration of an ototoxic compound, its consequences may be reversible or irreversible. So far, the cellular and molecular mechanisms underlying ototoxicity have only been deciphered for a few families of compounds. They remain indeterminate in most cases of ototoxicity, whether food- or drug-induced. Compounds for which ototoxic properties have been described include antibiotics, such as gentamicin, streptomycin, and tobramycin; diuretics such as furosemide; platinum-based chemotherapy agents such as cisplatin, carboplatin, and vincristine; as well as other chemical compounds such as 3,3′-iminodipropionitrile (IDPN). Rodent models of ototoxic damage have been developed for several decades [16,17] (Figure 2a). These models have made it possible to better understand the kinetics and mechanisms underlying the phenomenon of ototoxicity. However, aminoglycoside ototoxicity models are not used in research and development programs to test protective or regenerative approaches, due to the lack of reproducibility resulting from different interindividual resistances. It should be noted that a transtympanic injection model of cisplatin has recently been developed with the aim of circumventing the side effects of this compound on the kidney, which by generating severe pain, impairs monitoring of posturo-locomotor deficits linked to vestibular lesions themselves [18].

Transtympanic administration of arsanilate, an arsenic derivative, causes progressive and irreversible destruction of the vestibular sensory epithelia within a few days [19]. Although the neurotoxic mechanisms underlying this ototoxicity have not yet been elucidated, this model can be related to vestibular damage such as AUPV, labyrinthitis, vestibular aging (progressive loss of sensory cells, primary neurons or their synaptic contacts) or even high-grade vestibular schwanoma (benign tumor of the myelin sheath of the cochleo-vestibular nerve with progressive development). When carried out unilaterally, this chemical destruction generates in animals a vestibular syndrome characteristic of the unilateral peripheral vestibulopathies encountered in humans, associating posturo-locomotor and vestibulo-ocular deficits. Simultaneous or sequential bilateral administration of arsanilate makes it possible to reproduce bilateral vestibular areflexia reproducing those encountered in humans following massive administration of aminoglycosides or after heavy chemotherapy. Under these conditions, the animals also exhibit cognitive deficits of memory loss and spatial navigation [20,21,22] characteristic of those observed in patients with double vestibular areflexia [23].

#### 2.1.3. Models of Ischemic Damage

Situations of local ischemia in the inner ear are postulated to be the cause of vestibular disorders such as AUPV, labyrinthitis, or Menière disease [24,25,26]. Such situations of ischemia could result from stenosis or thrombosis of the cerebral arteries (AICA, PICA) which vascularize the inner ear organs [27]. Animal models of inner ear-induced ischemia have been developed in rodents and studies of resistance to ischemia have been carried out [28,29]. Although extremely valuable from a research point of view, this type of model is not really compatible with screening studies of pharmacological compounds because of the difficulty of setting it up. Other study models aimed at reproducing excitotoxic lesions (mainly through deafferentation of inner ear hair cells) resulting from the massive release of glutamate by the hair cells suffering from ischemia have also been developed (for review: [30]; and below: Models of sudden, partial and reversible unilateral vestibular areflexia).

#### 2.1.4. Endolymphatic Hydrops Models

The presence of swelling of the endolymphatic compartment was first described almost 80 years ago, in postmortem histopathology studies in Menière disease patients [31]. These swellings, which affect both the cochlear and vestibular compartments, cause at advanced stages complete destruction of the sensory organs. Today, validation by anatomical cerebral imaging (MRI) of their presence in the inner ear is one of the tools routinely used in the diagnosis of Menière disease [32]. Although hydrops is in a majority of cases associated with the symptoms of Menière disease, we still do not know how it appears, and how its physical characteristics affect the principle of hair cells mechano-electrical transduction at the base of the generation of auditory and vestibular sensory information. It is also not known how the hydrops can be controlled in order to reduce the associated symptoms and protect the inner ear sensory organs (for review [33]). Different animal models have been developed over the last 30 years in an attempt to reproduce both the functional consequences and the tissue damage associated with the endolymphatic hydrops. These models all have as common denominator a modulation of the endolymph ionic homeostasis [34]. The injection of artificial endolymph into the scala media, to increase the endolymph volume, or the surgical removal of the endolymphatic sac, in order to block its reabsorption, makes it possible to generate endolymphatic hydrops. In the same way, endolymphatic hydrops can be generated through fluid retention induced by antidiuretic hormones. This is the case with the administration of aldosterone, which induces hydrops by reabsorption of Na+ and Cl- through its labyrinthine receptors. Similarly, the administration of vasopressin acts on the V2-aquaporin 2 receptor couple present in the endolymphatic sac to generate hydrops. Although these models do not always induce auditory or vestibular deficits in animals, they nevertheless allow confirmation that local administrations of compounds likely to act on the osmolarity of the endolymph, such as diuretics, are capable of significantly reducing hydrops [34,35,36]. The degree of predictivity of these study models must however be balanced by the fact that the etiology of Menière disease is certainly multiple, and that hydrops is perhaps a histological correlate which is not mandatory to the expression of the vestibular syndrome. Interestingly, modulation of vestibular nuclei neurons by vasopressin through V1b receptors potentially contributes to the development of motion sickness in rats [37].

#### 2.1.5. Blast Models and Sound Trauma

A systematic review published by Stewart et al. pointed out the interest of animal models in researching the consequences of exposure to continuous and intense noise on the peripheral and central vestibular system (i.e., an increase in glutamic acid in the brainstem vestibular nuclei and in the cerebellum has been observed in rats 6 months after sound trauma) [38]. A physical inner ear trauma model has also been recently developed to reproduce the inner ear damage experienced during very loud sound or explosion stimulations [39]. This model demonstrated that these traumatic conditions triggered the generation of endolymphatic hydrops. Because of the technical constraints specific to the realization of the blast, this study model has not yet been used for research on the effects of active pharmacological compounds on the generation of hydrops or for the inner ear protection.

### 2.2. Animal Models Reproducing the Neurophysiological Processes Underlying the Vestibular Syndrome

Animal models of surgical, excitotoxic, or chemical lesions have been developed (for review [1]), in order to mimic the pathophysiological processes encountered in humans. They can be classified according to their impact on the vestibule function.

#### 2.2.1. Models of Sudden, Total, and Irreversible Unilateral Vestibular Areflexia

The unilateral surgical section of the vestibular nerve (unilateral vestibular neurectomy—UVN) mimics a sudden, total, and irreversible deafferentation of the vestibular sensors, as encountered in certain cases of AUPV or labyrinthitis. It reproduces identically the neurotomy procedure used in Menière disease patients refractory to pharmacological treatments and its clinical consequences [40]. It causes static and dynamic deficits as seen in patients with AUPV. The rodent model of UVN (Figure 2b) has recently made it possible to characterize the static [41] and dynamic [42] deficits associated with this type of impairment, through behavioral studies carried out with specifically adapted experimental devices (see the section below titled New Technological Approaches Available for the Behavioral Assessment of Vestibular Syndrome in Animals). It has also allowed the study of a set of cellular and molecular reactive mechanisms that occur at the level of the brainstem vestibular nuclei and which contribute to the phenomenon of central compensation, such as neurogenesis, gliogenesis, and neuroinflammation [43,44,45]. The vestibular syndrome generated by UVN reaches a maximum level of severity and is very reproducible, facilitating the study of the antivertigo properties of compounds of interest. The combination of these models and methods has made it possible to detect significant benefits of reference antivertigo compounds prescribed in human clinics and to identify their mechanism and sites of action [46,47,48,49].

The surgical labyrinthectomy model differs from UVN in that the destruction is confined to the peripheral vestibular receptors only, without cutting the VIII nerve (Figure 2c). The aim of this approach is to preserve the Scarpa ganglion, containing the cell bodies of the afferent vestibular fibers, in order to approach the pathogenic conditions postulated in certain AUPV, labyrinthitis, or in Menière disease [1]. This generally results in a partial preservation of the fibers of the vestibular nerve, although disconnected from their peripheral targets. Under these conditions of preservation of all or part of the primary neurons, the vestibular syndrome is close to that of UVN, although the kinetics of functional recovery is faster [50]. At the cellular level, the reactive mechanisms expressed in the vestibular nuclei after surgical labyrinthectomy are also less intense. Thus, it appears that the vestibular compensation mechanisms are closely dependent on the etiology of the disorder as previously mentioned [1]. Preserving the resting activity of deafferented primary neurons could thus allow faster recovery of electrophysiological homeostasis within the homologous vestibular nuclei. Labyrinthectomy is closer to a gradual process of aging of peripheral vestibular receptors, their injury by head trauma, or their poisoning by ototoxic compounds.

#### 2.2.2. Models of Sudden, Partial, and Reversible Unilateral Vestibular Areflexia (TTK)

The contact area between vestibular hair cells and afferent fibers forming the 8th cranial nerve is considered to be the most vulnerable area of the inner ear [51]. In the cochlea, selective damage to primary auditory synapses are implicated in acquired sensorineural hearing loss. A recent quantitative histopathological analysis of the inner ear in elderly subjects confirmed previous observations made in animals [52]. It indicates that cochlear synaptopathy and degeneration of peripheral cochlear nerve axons (despite nearly normal hair cell populations) may be essential components of age-related hearing loss. A few histopathological studies of human temporal bones have demonstrated segmental loss of isolated branches of the peripheral vestibular nerve distally to Scarpa’s ganglion, with or without degeneration of the associated sensory epithelium in patients with AUPV [53,54]. On the basis of these observations, it has been proposed that selective deafferentation of vestibular sensory cells may be involved in certain peripheral vestibulopathies such as vestibular neuritis, labyrinthitis, vertigo of ischemic origin, and Menière disease [25,54,55]. More recently, histological studies carried out on aged mice have shown a selective loss of peristriolar vestibular calyx synapses [56]. These targeted synaptic losses could be directly involved in age-related vestibular deficits.

The development of animal models of inner ear excitotoxic injury is aimed at reproducing the excitotoxic consequences of injury and damage to the sensory hair cells (Figure 3). The massive release of glutamate is in fact the consequence of the depolarization of hair cells in situations of ischemia [57]. Different types of models have been developed to deliver glutamate receptor agonists to the vestibule [58,59]. Histological and functional approaches have demonstrated that the induced hearing losses or vestibular deficits result from the deafferentation of sensory cells after swelling and detachment of auditory and vestibular nerve endings. Transtympanic administration of kainic acid (TTK), a glutamate receptor agonist, causes transient deafferentation of hair cells, accompanied by characteristic posturo-locomotor and vestibulo-ocular deficits [30,60,61,62]. The vestibular syndrome generated by TTK administration reproduces the static and dynamic deficits encountered in the AUPV patients, as well as their kinetics [30]. The TTK model shows in turn spontaneous reafferentation of vestibular sensors by the deafferented fibers of the vestibular nerve and allows screening of drug candidates that potentially stimulate neuronal regrowth and repair of peripheral synapses [30].

#### 2.2.3. Models of Peripheral Vestibular Neuroinflammation

The neuroinflammatory hypothesis in the context of AUPV of the vestibular neuritis type has been proposed on the basis of the suspicion of inflammatory foci along the different branches of the vestibular nerve [54,63,64]. With the development of vestibular functional assessment methods, questions regarding the precise anatomical areas of lesions that cause AUPV symptoms are increasingly raised. The hypothesis of an intralabyrinthine origin is now favored [65]. A few animal models of inner ear inflammation based on cytomegalovirus infection, or intratympanic lipopolysaccharide administration, have been developed for research purposes [66,67] (Figure 2a). The lack of reproducibility of these models and the fact that they do not systematically generate vestibular syndrome has so far precluded their use for screening compounds for therapeutic purposes.

#### 2.2.4. Models of Central Vestibular Inflammation

The UVN model generates central vestibular inflammation that essentially affects the vestibular nuclei disconnected from 8th nerve [50,68,69,70,71,72,73] (Figure 4). This model systematically reproduces the typical behavioral phenotype of an AUPV. It represents a model of interest for studying the role of central inflammation in vestibular physiopathology, but also the effect of pharmacological modulation of inflammation on the expression of the pathology [71,72]. Other vestibulopathy models are known to generate central vestibular inflammation, such as the arsanilate model [74], and the surgical labyrinth destruction model [50,75,76,77]. It can be mentioned that the induced central inflammation is the most intense in the UVN model.

#### 2.2.5. Models of Transient Chemical Blockade of Peripheral Neuronal Excitability: Vestibular Anesthesia

The principle of vestibular-suppression or vestibular-modulation consists in modulating the molecular effectors expressed throughout the vestibular sensory network in order to control the sensory information generated at the level of the vestibule and transmitted to the vestibular nuclei through the vestibular nerve. This approach, the origin of which dates back to the work of Barany in the first part of the 20th century [78], was based on the principle that the episodes of vertigo attacks observed in Menière disease patients resulted from a transient unilateral hyperexcitability of primary vestibular neurons, itself due to an unknown malfunction within a vestibule. The logical countermeasure then proposed by Barany consisted in counteracting this nervous hyperexcitability by means of neuronal activity blockers. To this end, he administered intravenously to Menière disease patients undergoing an acute crisis antagonists of voltage-sensitive sodium channels such as lidocaine. This operation, defined as vestibulo-suppressive or vestibulo-plegic, subsequently evolved for obvious reasons of cardiac and neurological risks towards a local application via the transtympanic route and took the name of “vestibular anesthesia” or “internal ear anesthesia” (for review [79]). Since this original work, many clinical studies have demonstrated a significant benefit of this type of local approach without this protocol becoming the standard treatment for acute episodes of Menière disease or other vestibular disorders such as those grouped by Halmagyi under the unilateral vestibular deafferentation syndromes [55,80,81,82]. This can probably be explained by the risks of inappropriate vestibular suppressant actions, i.e., carried out outside the time window of unilateral hyperexcitability or especially too pronounced unilateral inhibitions leading to an amplification of the imbalance of vestibular activity and what results in an exacerbation of the vertigo syndrome rather than its alleviation. 

Transtympanic administration of tetrodotoxin (TTX) proceeds from the same mechanism of blocking the excitability of vestibular cells. As the only ones to possess voltage-activated sodium channels, primary vestibular neurons are the main targets of the inhibitory action of TTX [58]. This operation causes the generation of a transient acute vestibular syndrome [50,83] (Figure 5). It is also a form of reversible functional deafferentation like the TTK model. 

#### 2.2.6. Models of Caloric Irrigation of the Inner Ear

Bárány was the first to understand that hot and cold irrigation of the auditory canal created convective movements of the endolymphatic fluid in the semicircular canals and thus activated the mechanoreceptor cells of the inner ear. With cold water being a vestibular depressant and hot water being a vestibular excitant. The caloric stimulation method, which earned him the Nobel Prize in 1914 [84], is now a commonly used tool for testing canal functions in humans and animals. One of its main advantages is to be able to induce a vertigo syndrome in a simple and non-invasive way. This method has made it possible in animals, among other things, to highlight the role of serotonin in inducing vertigo attacks [85] (Figure 6). It has also recently been used in humans, to validate the antivertigo potential of drug candidates [86].

## 3. New Technological Approaches Available for the Behavioral Assessment of Vestibular Syndrome in Animals

### 3.1. Subjective Quantitative Analysis

The assessment of the behavioral consequences of unilateral vestibular damage in rodent models is essentially based on a subjective quantitative assessment of a set of vestibular symptoms (circling, head tilt, falls, retropulsion, etc.), leading to varied score scales [41,44,60,62,87,88]. These different scales describe the overall kinetics of the syndrome where the symptoms are expressed at their peak the first three days post-injury (critical period) then gradually attenuate to return to normal values (vestibular compensation).

In the human model as in the animal model, a subjective quantitative analysis consists in assessing a certain number of physiological and/or posturo-locomotor markers (i.e., heart rate, sleep rhythm, step length, stance phase time, etc.), selected after a clinical phase of observation, in a known environment, in order to determine whether their variations, at a predetermined instant and/or over a defined period of time, are statistically significant. The ideal is to be able to collect this information with a minimum of interaction or with perfect control of the individual’s environment. The use of the animal model is therefore suitable for collecting data with a minimum of bias.

### 3.2. Automated Analysis: Dynamic Vestibular Parameters

The open-field animal video tracking test (Ethovision XT 14, Noldus) has been recently adapted to the animal models of peripheral vesibulopathy in order to identify and quantify in an automated and unbiased way different posturo-locomotor markers specific to AUPV. This analysis is carried out in dynamic and spontaneous conditions without any constraint for the animal. These markers are in some cases quantified for the first time. The behavioral analysis is able to provide data such as: locomotion velocity, distance traveled, quality of locomotion (meander), animal immobility time, acceleration of the locomotor pattern, body position in space, as well as the rotational behavior typical of vestibular damage [42].

### 3.3. Automated Analysis: Static Vestibular Parameters

A new method for automated analysis of the vestibular syndrome using the weight distribution device (DWB^®^, Bioseb, Vitrolles, France) has just been implemented. The DWB^®^ device, used for the first time, makes it possible to quantify new biomarkers of postural instability in the rodent model of AUPV. These parameters include: the surface of the sustentation polygon, the weight distribution of the animal on the lateral axis, the time of use of the abdomen as a support, the number of rotations carried out per unit of time during the phases of “circling”, and parameters similar to those used in clinical posturology, such as the barycenter [41,89].

### 3.4. Cognitive Analysis: Quantification of Cognitive Deficits

Spatial cognition is based on the integration of two types of sensory information: those called allothetic (based mainly on visual information) and those referred to as idiothetic (of which the vestibular system is a major source in mammals) [83]. Over the past two decades, it has become apparent that the vestibular system is involved in much more than just a reflex function. Numerous studies in animals and humans have demonstrated that vestibular dysfunction is associated with various forms of cognitive impairment, particularly related to spatial memory [90,91,92]. Many studies have explored the links between the vestibular system and spatial cognition in rodents. In particular, many have focused on the impact of vestibular lesions on navigational behavior in rats. In the case of bilateral vestibular lesions (peripheral destruction of vestibular receptors), behavioral deficits have been observed over the very long term, suggesting their potential irreversibility [93,94]. Regarding unilateral vestibular lesions, although deficits are observed for a very long time, they are not permanent [95].

## 4. How Can the Predictability of Peripheral Vestibulopathy Study Models Be Improved in the Future?

How can we best approach the indicators of human pathology? This is the great debate that has excited great interest among researchers who have been interacting with clinicians and rehabilitation practitioners for many years [96]. The authors of this review have already mentioned the impact of the experimental approach, making the self-evident observation that the translational process from the laboratory to the clinic is not exclusive. Observation carried out in the clinic, impacting experimental research on animals, is an essential process allowing, by means of the collection of patient information, the development of new approaches to clinical and therapeutic assessment. This is only possible by having the means to observe, collect, analyze and transfer this data to research laboratories. The case of the development of betahistine and tanganil^®^ illustrates the long-term partnership between basic and clinical studies which has shed light on the sites and mechanisms of action of betahistine, as well as of the active ingredient of tanganil^®^ [49,97], and on the other hand has enabled improvement of the posology of these drugs. In addition, in France the creation of the GDR (groupement de recherche) vertige (http://gdrvertige.com; 1 January 2015) offers the means to connect the various professionals working on vertigo and it has become clear that among the clinical data resource bases, in addition to medical data, are those held by physiotherapists who have developed their own rehabilitation clinic in the field [98]. This work has made it possible to establish an overview of the non-drug treatments proposed for the care of patients [40] and to begin a transfer to animals [73]. The long-term objective is to propose more standardized therapeutic approaches in order to optimize the functional results obtained, in order to better understand the underlying adaptive mechanisms and the consequences of protocolized therapies in the long term.

Improving the predictability of a study model requires getting as close as possible to the conditions of human pathology. When these conditions are not known, or poorly understood as in the case of most vestibular disorders, it will be a question of reproducing the symptoms of the pathology through approaches that mimic the neurophysiological processes at the base of the vertigo syndrome. Certain vestibular pathologies constitute in this sense a real challenge. This is the case, for example, of postural perceptual persistent vertigo (PPPD), a new entity that has led us to focus more specifically on the perceptual and emotional components. PPPD is most often manifested by instabilities, exacerbated by upright posture, active or passive movements, and complex visual stimuli. They are often accompanied by a significant state of anxiety [99]. These conditions are particularly difficult to assess in animals in the absence of verbal feedback. In this specific case, it will be a question of imagining tests that reflect the emotional and perceptual state of the animal. Although it is possible to transpose to animal models of vestibular disorders tests capable of quantifying the level of anxiety (e.g., raised cross maze, thygmotaxis in open field, etc.), there are still very few tests in the literature with a focus on purely perceptual aspects. An effort should be made on this point, in order to better understand the conditioning mechanisms of anxiety in the event of vestibular damage.

Biological biomarkers are too rarely used in vertigo clinics. However, many clinical studies indicate that specific hormonal profiles can promote the appearance of vestibular disorders, and that the expression of the vertigo syndrome is accompanied by measurable hormonal variations. The link between endocrine deregulation and vestibular alterations therefore no longer needs to be proven. However, there are still few data on the precise correlation of hormonal variations with the vertigo syndrome and very few studies have been devoted to the search for biomarkers allowing one to distinguish the different types and stages of vestibular pathologies [100]. Here again, a research and development effort is necessary in order to develop novel and more reliable diagnostic tools.

## 5. Conclusions

Given the lack of data on the etiopathogeny of peripheral vestibulopathies, and considering their inter-individual heterogeneity, animal models of vestibular pathologies offer the opportunity to develop new, more targeted and effective drug candidates to counteract the symptoms of PV. They are also particularly well suited to the analysis of the structural changes that occur in the vestibular pathway during rehabilitation approaches. The application to rodent models of new behavioral methods of postural-locomotor analysis compatible with the constraints of the human clinic (idea of tests at home) should allow further improvements in the diagnosis of peripheral vestibular disorders and help to define new biological, physiological, and behavioral markers for the long-term follow-up of patients.

## Figures and Tables

**Figure 1 biomedicines-10-03097-f001:**
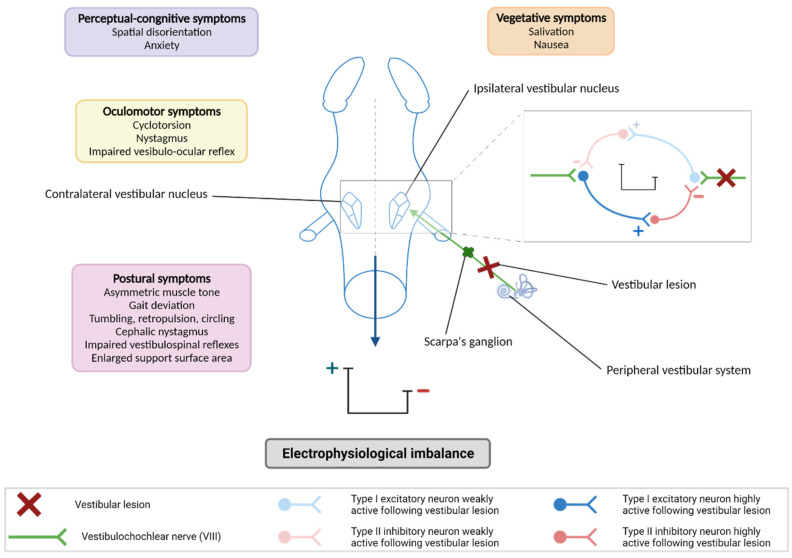
Neurophysiological basis of the vestibular syndrome. A vestibular lesion leads to an electrophysiological imbalance between the bilateral vestibular nuclei (VN) which is responsible for the vestibular syndrome. After a vestibular lesion, the ipsilateral vestibular nuclei are deafferented and show an excitability deficit in contrast to the contralateral vestibular nuclei to the lesion. High excitability and excitability deficit are represented on the scale (+ and – respectively). This effect is explained by an inhibition deficit of bilateral VNs: the excitatory neurons of type I present in the vestibular nuclei receive a different excitation from the vestibular primary neurons via the vestibular nerve depending on the side considered (lesioned or not lesioned), leading to a different activation of type II inhibitory neurons. The syndrome generated by the electrophysiological imbalance between the vestibular nuclei is composed of postural, locomotor, oculomotor, vegetative and perceptual–cognitive symptoms. Created with Biorender.

**Figure 2 biomedicines-10-03097-f002:**
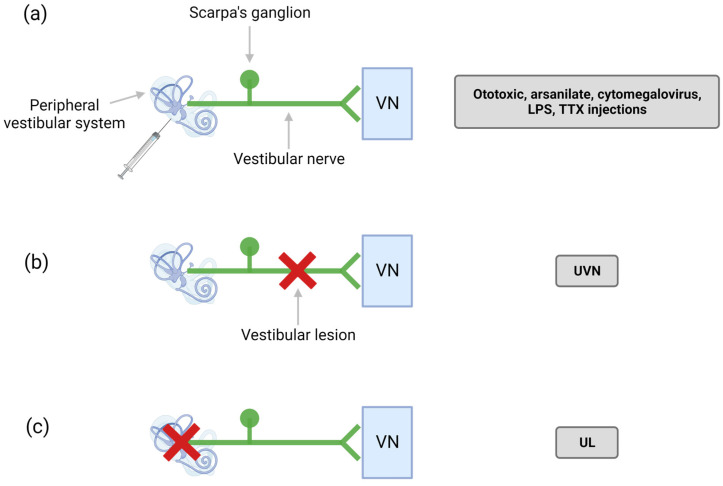
Models of vestibular lesions. Different models have been developed to study the vestibular syndrome. (**a**): injections of ototoxic compounds (antibiotics, chemicals, and chemotherapy agents), transtympanic injection of arsanilate, cytomegalovirus infection, lipopolysaccharide (LPS) injection and tetrodotoxin (TTX) injections. (**b**): surgical section of the vestibular nerve destroying the Scarpa’s ganglion referred to as unilateral vestibular neurectomy (UVN). (**c**): destruction of the vestibular receptors and preservation of the Scarpa’s ganglion referred to as surgical labyrinthectomy (UL). These models all induce an electrophysiological imbalance responsible for the vestibular syndrome. VN: vestibular nuclei. Created with Biorender.com.

**Figure 3 biomedicines-10-03097-f003:**
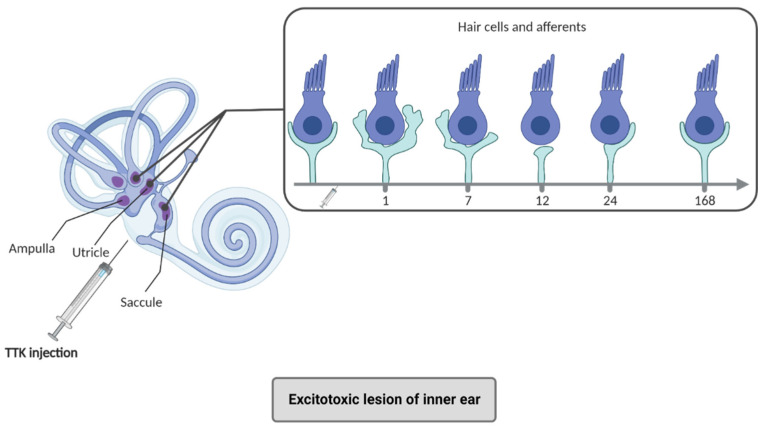
Excitotoxic lesion of inner ear and alteration of hair cells. Animal models of excitotoxic lesions of the inner ear such as TTK injection lead to suffering in the primary neuron terminals located in the otolithic organs (utricle, saccule) and ampullae according to a precise time course. At 1 h after the injection, the afferents inflate, leading to terminal resorption (7–12 h) and repair (24–168 h) (adapted from [60]). The contact zone between vestibular hair cells of the inner ear and the fibers forming the 8th cranial nerve is the most vulnerable zone of the inner ear and its alteration leads to a reversible vestibular syndrome. Created with Biorender.

**Figure 4 biomedicines-10-03097-f004:**
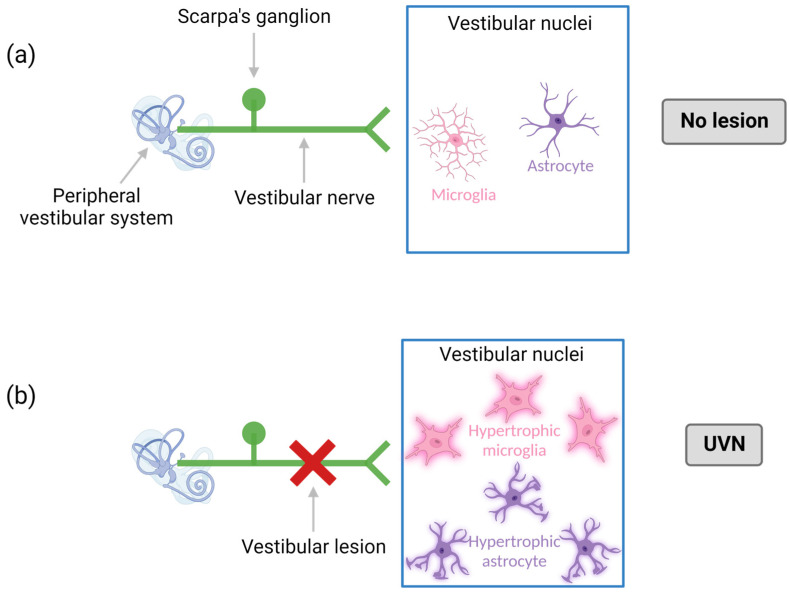
UVN as a central neuroinflammation model. The unilateral vestibular neurectomy (UVN) leads to an increase in microglial and astrocytic cells mainly in the vestibular nuclei, with hypertrophic phenotypes. Thus, UVN is a model of central inflammation. (**a**): without lesion, absence of inflammatory response. (**b**): with UVN, presence of hypertrophic microglia and astrocytes. Created with Biorender.com.

**Figure 5 biomedicines-10-03097-f005:**
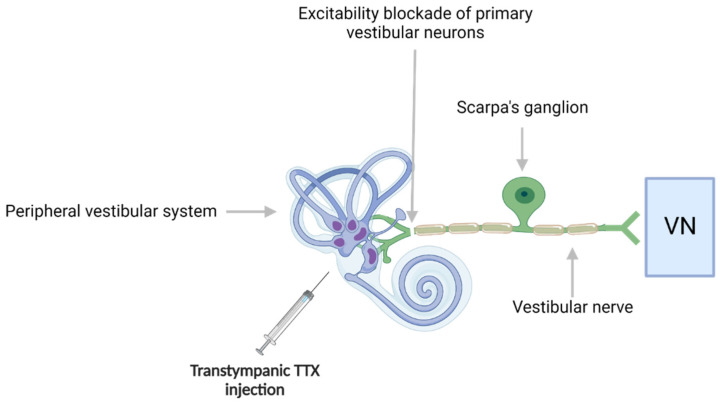
TTX injections and primary vestibular neurons. Transtympanic injection of tetrodotoxin (TTX) blocks the excitability mechanism in the primary vestibular neurons, leading to an electrophysiological imbalance which evokes a transient vestibular syndrome. Created with Biorender.com.

**Figure 6 biomedicines-10-03097-f006:**
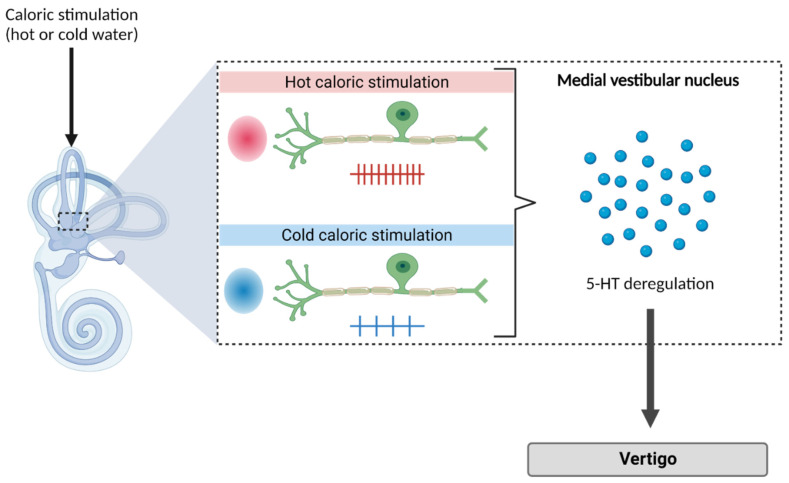
Serotonin role in the onset of vertigo. Caloric stimulation (hot or cold water) induces changes in primary vestibular afferent activity. Moreover, the immediate and significant increase in MVN 5-HT release is specifically the result of caloric stimulation. Thus, these increases would disrupt electrophysiological homeostasis and would participate in the potentiation of vertigo. Created with Biorender.com.

## Data Availability

Not applicable.

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
