# Peer review of "What Predictability for Animal Models of Peripheral Vestibular Disorders?"

_biomedicines, 2022, doi:10.3390/biomedicines10123097_

Round 1

Reviewer 1 Report

Dear Ladies and Gentlemen, Dear Journal-Team,

the manuscript 'What predictibility for animal models of peripheral vestibular disorders' lists and explains the various mechanisms that can induce vertigo. It is well written. The figures are sufficient.

a) Please explain the abbreviation TTX (tetrotodoxin) in Figure 2 as well, TTK (transtympanic kainic acid) in line 284, GDR (groupement de recherche) in line 441. 

b) Use uniform terminology for Meniere Disease, line 161, line 227, line 272, line 332, line 336, line 343.

c) Why didn't you use the term anxiety in Figure 1?

d) Please check the References for accuracy according to the Journal Style Guidelines. Check for unifrom capital letter use in Reference 1 (Lacour et al.). Name the language the article is written in at the end of the reference, when it is not in English, and give if possible an English translation (63, Ruttin, 78, Barany).

e) Language: 1. Line 75, please change to: 'an unilateral'.

2. Line 167, change to: 'Other study models aimed to reproduce...by the hair cells that ischemic suffering has'.

3. Line 348, change to: 'of vestibular activity and what results in an'.

Sincerely,

Author Response

a) Please explain the abbreviation TTX (tetrotodoxin) in Figure 2 as well, TTK (transtympanic kainic acid) in line 284, GDR (groupement de recherche) in line 441. 

As required, we detailed the abbreviations at the different slots mentioned

b) Use uniform terminology for Meniere Disease, line 161, line 227, line 272, line 332, line 336, line 343.

The term Menière disease has been corrected at all places mentioned.

c) Why didn't you use the term anxiety in Figure 1?

The term anxiety has been corrected in Fig.1

d) Please check the References for accuracy according to the Journal Style Guidelines. Check for uniform capital letter use in Reference 1 (Lacour et al.). Name the language the article is written in at the end of the reference, when it is not in English, and give if possible, an English translation (63, Ruttin, 78, Barany).

Corrected

e) Language: 1. Line 75, please change to: 'an unilateral'.

Done

Line 167, change to: 'Other study models aimed to reproduce...by the hair cells that ischemic suffering has'.

Done

Line 348, change to: 'of vestibular activity and what results in an'.

Done

Additional corrections:

The balance drawing has corrected in Fig1

Reviewer 2 Report

What predictability for animal models of peripheral vestibular  disorders?

The topic is interesting, this paper is well written and a great work has been done

I have only few remarks.

Methods

It could be interesting to  know which data based have been screened and the period of publication of the papers included in this review.

2.2.5. Models of transient chemical blockade of neuronal excitability: vestibular anesthesia

 This section has, in my opinion , too much space for Barany experiments and less to more recent experiences

2.2.6. Models of caloric irrigation of the inner ear

These both  sections could be better clarified  for the readers, especially the models of  caloric irrigation of the inner ear.

3. New technological approaches available for the behavioral assessment of vestibular  syndrome in animals

In my opinion,to avoid a self-promotion, it  should be used a verb in third person when you present the models you have implemented

Lines 419-420“ Many studies have explored the links between the vestibular system and spatial cognition in rodents. In particular, many have focused on  the impact of vestibular lesions on navigational behavior in rats”. I would expect some references….

Is it appropriate  to write “Cognitive deficits” concerning animal models…? In my native language the  term  “cognitive” is used only for humans..   

Author Response

Methods: It could be interesting to know which data based have been screened and the period of publication of the papers included in this review.

The references chosen to illustrate the different animal models of vestibular pathology were selected on the basis of their relevance to the notion of predictability of study models, rather than on a systematic classification from a bibliographic database. We also prioritized the study models that have become the most commonly used reference models in vestibular pathophysiology research laboratories.

2.2.5. Models of transient chemical blockade of neuronal excitability: vestibular anesthesia: This section has, in my opinion, too much space for Barany experiments and less to more recent experiences.

We have chosen to highlight Barany's original work in the approach to blocking vestibular nerve excitability. Other chemical compounds such as histamine receptor modulators are potential modulators of vestibular primary neuron excitability (and have provided the basis for many antivertigo drugs), however, they are not used (to our knowledge) as pharmacological tools to generate animal models of vestibular pathologies. Therefore, they are not cited in this review.

2.2.6. Models of caloric irrigation of the inner ear: These both sections could be better clarified  for the readers, especially the models of caloric irrigation of the inner ear.

We added a sentence in L363-364 to clarify the action of water irrigation.

  1. New technological approaches available for the behavioral assessment of vestibular  syndrome in animals: In my opinion, to avoid a self-promotion, it  should be used a verb in third person when you present the models you have implemented.

Corrected

Lines 419-420“ Many studies have explored the links between the vestibular system and spatial cognition in rodents. In particular, many have focused on  the impact of vestibular lesions on navigational behavior in rats”. I would expect some references….

References related to this specific sentence (93-95) are listed in the two following sentences.

Is it appropriate to write “Cognitive deficits” concerning animal models…? In my native language the  term “cognitive” is used only for humans.

The term "cognitive deficits" is widely used in the literature concerning animal models and more specifically rodents.